# Polysaccharides and Metal Nanoparticles for Functional Textiles: A Review

**DOI:** 10.3390/nano12061006

**Published:** 2022-03-18

**Authors:** Marta Fernandes, Jorge Padrão, Ana I. Ribeiro, Rui D. V. Fernandes, Liliana Melro, Talita Nicolau, Behnaz Mehravani, Cátia Alves, Rui Rodrigues, Andrea Zille

**Affiliations:** Centre for Textile Science and Technology (2C2T), Campus de Azurém, Universidade do Minho, 4800-058 Guimarães, Portugal; marta.fernandes@det.uminho.pt (M.F.); afr@2c2t.uminho.pt (A.I.R.); ruidvfernandes@gmail.com (R.D.V.F.); liliana.melro@2c2t.uminho.pt (L.M.); tali_nicolau@hotmail.com (T.N.); behnaz.mehravani@yahoo.com (B.M.); catia.alves.98@hotmail.com (C.A.); ruipvrodrigues@hotmail.com (R.R.); azille@2c2t.uminho.pt (A.Z.)

**Keywords:** functional textiles, polysaccharides, metal nanoparticles, chitosan, alginate, starch, cyclodextrin, cellulose

## Abstract

Nanotechnology is a powerful tool for engineering functional materials that has the potential to transform textiles into high-performance, value-added products. In recent years, there has been considerable interest in the development of functional textiles using metal nanoparticles (MNPs). The incorporation of MNPs in textiles allows for the obtention of multifunctional properties, such as ultraviolet (UV) protection, self-cleaning, and electrical conductivity, as well as antimicrobial, antistatic, antiwrinkle, and flame retardant properties, without compromising the inherent characteristics of the textile. Environmental sustainability is also one of the main motivations in development and innovation in the textile industry. Thus, the synthesis of MNPs using ecofriendly sources, such as polysaccharides, is of high importance. The main functions of polysaccharides in these processes are the reduction and stabilization of MNPs, as well as the adhesion of MNPs onto fabrics. This review covers the major research attempts to obtain textiles with different functional properties using polysaccharides and MNPs. The main polysaccharides reported include chitosan, alginate, starch, cyclodextrins, and cellulose, with silver, zinc, copper, and titanium being the most explored MNPs. The potential applications of these functionalized textiles are also reported, and they include healthcare (wound dressing, drug release), protection (antimicrobial activity, UV protection, flame retardant), and environmental remediation (catalysts).

## 1. Introduction

Functional textiles have triggered enormous interest in various industrial sectors, displaying an annual growth rate of 30% between 2015 and 2020. This growth has been mainly supported by the automotive, fashion, healthcare, military, and sports industries. Some examples of functional textiles include antimicrobial, insect-repellent, oil/water-repellent, odor-control, flame retardant, heat-resistant, windproof, sensing, ultraviolet (UV) protective, thermoregulating, antistatic, electrically conductive, antiwrinkle, self-cleaning, and conductive properties [1,2,3].

Nanotechnology has been widely used to prepare functional textiles and represents a new thread in clothing technology. Textile-based nanoproducts make use of all the advantages of nanomaterials, creating and tailoring high-value properties without hindering the comfort and flexibility of the substrates [4]. These materials are developed using conventional textiles functionalized with nanoparticles (NPs) or using textiles composed of nanofibers [5,6]. Metal nanoparticles (MNPs) in particular have been extensively studied for textile functionalization due to their distinct physicochemical and biological properties (Figure 1) [7,8]. MNPs play an important role in this technological evolution because they show outstanding surface properties that allow higher effects compared with bulky conventional additives [9,10]. Various metal and metal oxide NPs have been used, such as silver (Ag), gold (Au), copper/copper oxide (Cu/CuO), zinc/zinc oxide (Zn/ZnO), titanium/titanium dioxide (Ti/TiO_2_), nickel (Ni), cobalt (Co), and iron/iron oxide (Fe/Fe_3_O_4_/Fe_2_O_3_) [11].

Polysaccharides are highly relevant chemicals in the development of textile materials, as they can be used as building blocks of fibers, coatings, or stabilizing components [13,14,15,16,17]. They represent are renewable resources that have been largely studied due to their biocompatibility, biodegradability, and diverse biological activity (e.g., anti-inflammatory, immunoregulation, antitumor, anticoagulation, antioxidant, antimicrobial, and hypoglycemic activity) [18,19,20]. Therefore, they have been widely applied in materials engineering, namely in packaging systems, tissue engineering, controlled drug delivery, flexible electronics, and 3D printing [21,22,23,24,25]. They are natural macromolecules composed of monosaccharide units covalently linked by glycosidic bonds to form polymer chains [26,27]. Nowadays, thousands of polysaccharides are extracted from natural sources and can be obtained from plants (e.g., cellulose, starch, and pectin), algae (e.g., alginate, agar, fucoidan, and carrageenan), animals (e.g., hyaluronic acid, heparin, chitin, and chitosan), and microorganisms (e.g., xanthan gum, dextran, pullulan, and bacterial cellulose) (Figure 2) [28,29,30,31,32].

Textiles that incorporate both organic and inorganic constituents, or hybrid textiles, are an emerging and promising segment of the textile industry. Hybrid textiles may display a synergistic effect between their constituents and are therefore able to enhance their range of properties and activity, improving the final products. Numerous combinations are possible, leading to several novel textiles with unpaired properties [33]. Polysaccharides present several opportunities when combined with different MNPs. In the last decade, an increasing interest has been observed in the usage of polysaccharides derivatives and their nanosystems to generate hybrid materials. Novel and improved functionalizations allow for exploration and development of new areas of application that were previously impossible to achieve [34,35].

This review is focused on hybrid textiles that encompass polysaccharides and MNPs. Furthermore, the role of polysaccharides is highlighted. Considering textile hybrid materials, the use of polysaccharides can be useful in at least four different ways: (i) by its presence in the substrate composition; (ii) to enhance the adhesion of MNPs onto fabrics, acting as a binder and/or controlling the release of NPs or metal ions; (iii) to improve the multifunctional properties of textiles (e.g., antimicrobial, UV protective, self-cleaning, easy-care, and flame-retardant properties); and (iv) as reducing agent of metal salts and/or to stabilize MNPs on dispersions (Figure 3).

As depicted in Figure 3, polysaccharides host metal ions through noncovalent bonding (sorption). The as-formed metallic precursor is then reduced to a zero-valent state, starting nucleation and nanocrystal growth simply by altering the order of free energy (heating). The increase in temperature stabilizes the MNPs and allows for control of their morphology and growth kinetics. This type of self-assembly (bottom-up) synthesis is preferred over top-down synthesis, where the starting materials are reduced in size via mechanical, thermal, or chemical treatments. These treatments may induce an unwanted oxidation of the NPs and consequently alter their physical properties and/or surface chemistry. Furthermore, the stabilized MNPs do not easily leach the coordinated metal ion unless there is an external stimulus, such as a pH change. Most polysaccharides are susceptible to pH alteration and are therefore often used for controlled release and drug delivery in polysaccharide-based systems [36].

## 2. Polysaccharides in Metal-Nanoparticle-Functionalized Textiles

### 2.1. Chitosan

Chitosan is a type of polysaccharide that derives from the deacetylation of chitin, a homopolymer of β-(1–4) linked *N*-acetyl-D-glucosamine [37]. It has gained visibility because its raw component, chitin, is commonly found in nature either in invertebrate animals or in the cell walls of algae or fungi [38]. This biopolymer is a versatile material due to its abundant amino and hydroxyl groups, which provide highly reactive and interesting physicochemical properties [39,40]. Despite its hydrophobic backbone at high pH, for pH below its pKa (pH < 6.2), the amino groups (NH_2_) on the chitosan chains are protonated into the positively charged group (NH_3_^+^), enabling its solubility in water [41,42]. Chitosan has been considered a renewable, biocompatible, biodegradable, antioxidant, film-forming, antimicrobial, mechanical, and thermally stable material [39,43].

Chitosan has been applied in combination with MNPs in textile materials, comprising a multitude of sectors. These textiles have been indicated for tissue engineering, controlled delivery of pharmaceutical agents, wound dressings, smart garments, and packaging [44,45,46]. Works describing the use of MNPs and chitosan in textile applications are depicted in Table 1. Here, different fibers were found, such as cotton, polyamide, polyester, polypropylene, polyethylene, aramid, linen, ramie, viscose, chitosan, and alginate. In the MNPs component, mostly Ag, Cu, and Zn NPs were used, although materials containing Ti, Fe, Co, Ni, and cesium (Ce) NPs were also prepared.

In this section, the works were divided according to the chitosan role on hybrid materials. Chitosan was used in substrate composition, as well as to enhance the adhesion of MNPs onto fabrics and/or control the release of MNPs or metal ions. In other works, chitosan acted as a reducing agent of metal salts and/or a stabilizer of MNP dispersions. Finally, chitosans can also contribute to improving the physical and mechanical properties of textiles (e.g., tensile strength, as well as air and water permeability), as well as the biocompatibility of textile-based composites.

#### 2.1.1. Substrate Composition

Chitosan has been applied as a textile substrate to produce wearable electrodes, scaffolds for wounds, and catalysts for the removal of water pollutants. Chitosan substrates have been commercially obtained [16] via electrospinning [44,78] or wet spinning [46] methods.

Qin et al. developed novel wearable electrodes for smart garments by electroless plating of AgNPs onto the surface of commercial chitosan fibers. The developed electrode could maintain its electrical resistance, since the amine groups in the chitosan structure strongly react with metal ions due to their nitrogen atoms holding free-electron doublets, leading to interesting conductivity values, even after washings. To evaluate the proposed electrodes, the authors embedded the nanofibers into garments to capture electrocardiogram signals in various motions, showing good data-acquisition ability and sensitivity [16].

Ali et al. prepared different chitosan–TiO_2_ composites with different metal ions (Cu^2+^, Co^2+^, Ag^+^, and Ni^2+^) to obtain highly efficient and easily retrievable catalysts by wet spinning. Chitosan was selected due to the presence of reactive amino groups able to sorb metal ions by electrostatic and chemical interactions. TiO_2_NPs were added to improve the mechanical and chemical properties of chitosan, as well as to counter its weak mechanical strength and dissolution in acidic medium. After metal-ion loading, the composites were treated with sodium borohydride to obtain zero-valent MNPs. After the application of these MNPs, the authors observed an improvement in catalytic efficiency for the reduction of organic dyes and nitrophenols. Moreover, the composites could be easily recovered and reused [78].

In the biomedical sector, Ahmed et al. produced nanofiber mats for diabetic wounds combining chitosan, polyvinyl alcohol (PVA), and ZnONPs via electrospinning technique. The novel mats were shown to promote faster wound healing with reduced inflammatory exudate and considerably less necrotic material during the healing process due to the antibacterial activity of ZnONPs and the antioxidant properties of chitosan. Chitosan-based electrospun nanofiber mats have presented a huge potential for diabetic wound treatments because they resemble the extracellular matrix and promote cell adhesion, proliferation, and wound closure. The mats presented high antibacterial activity against *E. coli, P. aeruginosa*, *B. subtilis,* and *Staphylococcus aureus*, as well as antioxidant potential. The properties of the nanofiber mats were confirmed by in vivo experiments, representing an interesting application as wound dressing [45]. Ezzat et al. produced nanofibers by electrospinning PVA and carboxymethyl chitosan (CMCh) with AgNPs. The chitosan acted as a support and stabilizing agent for the NPs. The antimicrobial properties of the developed nanofibers were evaluated using the agar diffusion method against bacteria *(Klebsiella pneumoniae**, E. coli,* and *S. aureus)* and fungi (*Candida*
*albicans*). The nanofibers showed strong antibacterial activity against the tested strains and were presented as promising active wound-dressing materials [79].

#### 2.1.2. Enhancing the Adhesion of Metal Nanoparticles onto Textiles and/or Controlling the Release of Nanoparticles or Metal Ions

As mentioned previously, chitosan can be used to improve the adhesion of metal salts or NPs onto cotton, linen, polyamide, and aramid fabrics through molecular forces or a double network. The addition of chitosan has been shown to increase the washing fastness of the treated composites to obtain persistent antimicrobial properties or color resistance to fading after several washing cycles. Hasan et al. functionalized aramid fibers with chitosan and AgNPs without employing toxic reagents in the process. In this context, chitosan was used as a stabilizing agent. According to the authors, this functionalization created novel characteristics in terms of color and antibacterial activity against *E. coli* and *S. aureus* to aramid fibers, as well as improved thermal resistance. It should be highlighted that antibacterial activity maintained constant efficacy, even after 15 washing cycles [46]. Said et al. modified gauze fabrics with chitosan (cationization) or partial carboxymethylation (anionic modification), where these chemical modifications improved NP deposition [17]. Gadkari et al. modified a woven cotton fabric to provide antimicrobial properties through a layer-by-layer coating without damaging the physical and mechanical properties. To apply this technique, the authors used both anionic (polystyrene sulfonate, PSS) and cationic (synthesized Ag-loaded chitosan agents) auxiliaries [72]. Xu et al. prepared a cotton fabric with antibacterial properties by one-pot modification technique (pad–dry–cure) using CMCh as a binder and stabilizer. The dispersion of the AgNPs on the cotton fabric was uniform, owing to the presence of amine groups and ester bonds in the CMCh, which enables the formation of coordination bonds with AgNPs and hydroxyl groups of the cellulose. Later, the antibacterial properties of these fabrics were tested against *E. coli* and *S. aureus* and showed an impressive bacterial reduction, even after 50 laundering cycles (94% and 96% for *E. coli* and *S. aureus,* respectively) [77]. Hajimirzababa et al. coated sodium alginate microcapsules on the surface of cotton gauze, which were prepared via two methods. In the first method, AgNPs were formed inside of alginate microcapsules, and in the second method, AgNPs were formed on the outside of alginate microcapsules. Both types of microcapsules were loaded into the already-prepared polyvinylpyrrolidone-iodine (PVP-I) and chitosan solution, followed by impregnation on the cotton gauzes. In this study, chitosan controlled the release of AgNPs and was also used to improve the biocompatibility of the gauzes [61]. Xu et al. produced a nanocomposite coated with Ag/TiO_2_NPs. NP dispersion was prepared using CMCh as a stabilizing agent of the NPs. Then, the NPs were deposited on the fabric by the pad–dry–cure method. The heating condition during the deposition method (180 °C) leads to the reaction of the carboxyl group of CMCh chain with the hydroxyl group of the cotton cellulose via esterification, promoting a covalent grafting between CMCh and cotton fabric. Then, the adhesion of the NPs was also promoted due to the formation of covalent bonds between TiO_2_NPs and amino groups in CMCh. Likewise, CMCh formed coordination bonds with Ag [73]. Sheikh et al. developed a multifunctional textile wherein linen fabric was coated with in situ synthesized AgNPs using tamarind seed extract as a reducing agent of Ag ions. Then, a chitosan formulation was padded to serve as a template to immobilize the AgNPs onto linen. The samples with higher antibacterial activity were further tested for durability, and it was found that even after 50 washing cycles, there was no considerable reduction in antibacterial activity. The presence of a positive charge in chitosan and negatively charged tannin enhanced the adherence of the NPs onto the surface, promoting a proper attachment of the AgNPs. The presence of NPs, *N*-containing chitosan, and tannins also played a synergistic role in flame retardance [62].

#### 2.1.3. Multifunctional Textiles

In addition to the properties conferred by the NPs, chitosan by itself may improve several individual properties.

In the antimicrobial field, chitosan alone mainly presents a bacteriostatic effect, and different mechanisms of action have been proposed. The greatest consensus about chitosan antimicrobial action is related to the electrostatic interaction between the positive charge of chitosan with the negative surface charge of microorganisms that disrupt the stability of cell membranes/cell walls of pathogens. Furthermore, additional antimicrobial mechanisms have been proposed, namely chitosan interaction with the microbial genome, chelate nutrients, forming a thick film enveloping the microorganisms, thus limiting their accessibility to nutrients [80]. The combination of chitosan and MNPs can act in an additive or synergistic way to improve antimicrobial activity. Therefore, several researchers performed this conjugation onto textiles. These hybrid materials have been indicated for use in hygienic, healthcare, and packaging products. Together with antimicrobial activity, Ramadan et al. improved air permeability and water-absorbance properties of the fabrics [47].

Regarding UV protection, chitosan may have a positive effect due to their ability to form films on fabrics and provide superior adhesion of MNPs onto the surface of fabrics. In the latter case, MNPs easily absorb and scatter UV rays [48,81]. Similarly, self-cleaning and easy-care properties may be obtained for the same reasons. Here, MNPs have a crucial role in photocatalytic activity and in increasing hydrophobicity [82].

The improvement of flame-retardant properties using chitosan and MNPs has been attributed to the presence of nitrogen atoms in the chitosan structure, combined with thermally resistant MNPs [74]. Another strategy is the phosphorylation of chitosan to introduce phosphorus moieties as a flame retardant. In this case, phosphorus and MNPs were selected for flame-retardant properties, and chitosan and Ag were selected for antibacterial activity [54].

Cotton fabrics have been commonly functionalized with chitosan and MNPs by padding [47,48,50,55,60] and dipping [49] methods to obtain antimicrobial textiles.

Rehan et al. investigated the effect of chitosan, AgNPs, and clay on the antibacterial properties of cotton fabrics. AgNPs were prepared with the help of the photochemical reduction method, where the chitosan acted as a stabilizing agent and assisted the antimicrobial action of the composites. The durability of the functionalization was tested after 20 washing cycles, and the antimicrobial activity against *E. coli* was assessed. The antibacterial effect of composites was reduced to 61% for the composites containing just chitosan, 87% for those with chitosan-AgNPs, and 91% for the chitosan-AgNP-clay composites. All samples showed UV protection. When investigated for flame retardance, better values were found in chitosan-AgNP-clay composites [55]. Montaser et al. produced a self-cleaning and antibacterial composite. In this work, the influence of adding a polyvinyl acetate (PVAc) copolymer and TiO_2_NPs with and without Zn doping on the antibacterial activity of cotton fabric was studied. Four samples were tested: cotton fabric as control, cotton fabric grafted with PVAc, cotton fabric grafted with PVAc and TiO_2_NPs, and cotton fabric treated with PVAc and Zn-doped TiO_2_NPs. The polymer and polymer/MNPs were coated onto cotton fabric by the pad–dry–cure method. Antimicrobial tests were performed against *S. aureus, P. aeruginosa* bacteria, and fungus *Aspergillus niger*. Photocatalytic tests showed that the cotton fabric containing TiO_2_NPs had an improved photocatalyst effect compared with the other samples. Furthermore, no synergistic effect was observed by combining TiO_2_NPs and ZnONPs [50]. Zayed et al. proposed the use of *Psidium guava* leaf extracts as a reducing agent to synthesize ZnONPs, serving as a greener route relative to traditional reducing agents. Additionally, the authors functionalized cotton fabrics with biopolymers (alginate, chitosan, and carboxymethyl cellulose (CMC)). The ZnO-chitosan produced better results in terms of UV protection and antimicrobial activity against *S. aureus*, *E. coli*, and *C. albicans* in comparison to other treated fabrics [52]. Hatami et al. and Mogrovejo-Valdivia et al. functionalized polyester fabric with AgNPs and chitosan to obtain antimicrobial textiles. Hatami et al. attempted to prepare phosphorylated chitosan by heating chitosan in the presence of orthophosphoric acid, urea, and dimethylformamide before the addition of Ag nitrate solution. Later, films were prepared by solvent evaporation and subsequently melt-blended with polyester to form a composite by compression. A significant improvement was observed with the addition of AgNPs and phosphorylated chitosan, showing good resistance to bacterial growth. Flame-retardant properties were also improved with the addition of phosphorylated chitosan and AgNPs [54]. Mogrovejo-Valdivia et al. also produced a polyester dressing material where a non-woven material was treated with chitosan and cyclodextrin by crosslinking with citric acid in a pad–dry process before soaking in an Ag sulfate solution. Afterwards, the fabrics were subjected to heat treatment, enabling the formation of AgNPs. Subsequently, a polyelectrolyte multilayer film was deposited using a layer-by-layer process. The samples showed good antimicrobial results against *S. aureus* and *E. coli* [51].

#### 2.1.4. Action as a Reducing Agent of Metal Salts and Stabilization of Metal Nanoparticles on Dispersions

Another function of chitosan in composites included in this review is their activity as a reducing and stabilizing agent during MNP synthesis. Hasan et al. coated a polyamide fabric surface with AgNPs synthesized in situ using chitosan as a reducing and stabilizing agent. First, the fabrics were immersed into a metal salt solution that allowed for the absorption of metal ions onto the polyamide surface. Secondly, the chitosan solution was dispersed over the Ag-cation-absorbed polyamide surface to form a ternary complex. Lastly, this complex reacted with the OH ions from chitosan to form AgNPs. The fixation of AgNPs was achieved through the interaction of Ag ions with amino and hydroxyl groups of polyamide and chitosan. Chitosan was able to stabilize the produced particles by creating a thin layer of cladding over their surfaces. When tested for durability, the bacterial reduction was observed to be more than 88% after 20 washing cycles. Along with antibacterial properties, color strength (K/S) was investigated, and it was found that K/S values were improved [68]. Mokhena et al. prepared a composite for water filtration, where the barrier was produced by electrospinning of alginate, and subsequently, a layer of chitosan and AgNPs was added. The AgNPs were synthesized in the presence of chitosan through a thermal treatment. This polysaccharide acted as a reducing and capping agent. The coated membrane was studied for antibacterial properties, dye removal, and oil separation [69]. Hasan et al. attempted the synthesis of AgNPs using chitosan, also as a reducing and stabilizing agent. The dispersion was used to create a coating on polyester [70]. Raza et al. coated viscose fabric with AgNPs synthesized and stabilized using chitosan to improve the antibacterial performance. The samples showed strong antimicrobial efficacy against *E. coli* and *S. aureu**s* [71].

### 2.2. Alginate

Alginates are natural anionic polysaccharides that can be extracted from different species of brown sea algae or bacteria, such as *Pseudomonas* and *Azotobacter* [83]. Alginate polymers are constituted by two monomers, D-mannuronic acid (M blocks) and L-guluronic acid (G blocks), and linked by β-(1,4) (M residue) and α-(1,4) (L residue) glycosidic bonds, forming a copolymer that can have different types of blocks (MM, GG, or GM) depending on the extraction source [84]. Commercially, they can be found in different forms of salts (sodium, potassium, or ammonium) with different molecular weights and distributions of M and G blocks. All these different forms alter the polymer physicochemical properties, such as the capability to uptake water, viscosity, and sol/gel transitions. In addition, heterogeneous GM structures are water-soluble at lower pH when compared to GG- or MM-structured polymers. As their properties can be modulated depending on their physicochemical properties, alginates can be used in several different fields, such as the textile, pharmaceutical, biomedical, agriculture, and food and beverage industries. [83,84]. Alginates are biocompatible, biodegradable, and non-toxic and are used as a thickener, reducing agent, and stabilizer, as well as in film formation. Moreover, alginates allow for the formation of hydrogel beads, fibers, scaffolds, or coatings when crosslinked with divalent ions (Ca^2+^, Ba^2+^, Fe^2+^, Zn^2+^, Sr^2+^, Ni^2+^, Cu^2+^, Co^2+^, and Pb^2+^, among others), which form a chelating center between alginate polymeric chains, resulting in a three-dimensional network (egg-box structure). This gelation method can be advantageous, since it can be carried out at room temperature and neutral pH. A summary of works combining MNPs with alginate within textile applications is presented in Table 2.

Ag is often combined with alginate by substitution of other monovalent ions, such as Na^+^ or K^+^. Moreover, alginate can act as a reducing agent (Ag^+^ → Ag^0^) and stabilizer, preventing agglomeration in the synthesis of AgNPs due to the abundance of multifunctional groups in its structure [87]. AgNPs can be combined with alginate to be used as a coating/finishing agent in fabrics or they can be incorporated into the alginate fiber itself in wet or electrospinning processes. For example, Zhao et al. reported the wet spinning of alginate fibers embedded with AgNPs, where the antibacterial activity was proven efficient against both *E. coli* and *S. aureus* and had a high cell-killing efficiency in human cervical cancer (HeLa) cells [94]. Zhang et al. also produced alginate nanofibers incorporated with Ag via electrospinning, where Ag was reduced in situ to obtain a highly sensitive humidity sensor able to monitor human breath [15]. Other authors combined non-woven electrospun alginate with poly(ethylene terephthalate) (PET), which was coated with AgNP-loaded chitosan to produce a thin film composite membrane for water purification, presenting antibacterial activity against both Gram-positive and Gram-negative bacteria, as well as the capacity to remove up to 93% of oils and NP retention greater than 98% [69]. Hajimirzababa et al. also used a combination of polysaccharides, alginate microcapsules loaded with AgNPs, and a chitosan solution containing PVP-I in a cotton gauze for wound dressing with excellent antibacterial efficiency [61]. Ag compounds can be used as oxidants to prepare composites, as demonstrated by Ji et al., who obtained highly conductive, hydrophobic, and UV-resistant non-woven alginate fabrics by in situ preparation of a polypyrrole/Ag (PPy/Ag) composite on the fabric. Coated fabrics presented conductivities of up to 2.0 × 10^−2^ S·cm^−1^ and only minor changes after 60 h of UV irradiation and 168 h of water soaking. These fabrics also exhibit excellent antistatic properties and thermal stability [85].

As previously mentioned, alginates can reduce and stabilize AgNPs in their synthesis, making this chemical reaction more ecofriendly. This greener approach has been mentioned by Mahmud et al. in several papers, as they have used sodium alginate as reducer and stabilizer of AgNPs in the surface functionalization of silk [86], cotton [87], and ramie [88] fabrics. The objective was to obtain colored fabrics with improved washing fastness, which proved to be higher when compared to traditional dyeing processes. All fabrics presented antibacterial activity and UV-protection properties. Tensile properties and crease recovery angle were improved in silk fabrics, and ramie fabrics presented catalytic action in the reduction of 4-nitrophenol.

Besides AgNPs, ZnONPs and CuONPs are also used to provide fabrics with UV protection and antibacterial activity. Regarding works performed with ZnNPs, alginate can be used as: (i) a stabilizer in the synthesis of ZnONPs, which were used to functionalize cotton fabrics [90]; (ii) a matrix to entrap ZnONPs, which were coated via a pad–dry–cure process onto cotton fabrics [52]; (iii) or as an ion exchanger in the in situ synthesis of ZnONPs in calcium alginate fabrics [89]. As for works regarding CuNPs, authors Marković et al. reported the use of sodium alginate as an ion exchanger in the production of antimicrobial composites in plasma-activated polypropylene non-woven fabric [91] and in polyester and polyamide fabrics [92]; all three fabrics showed antimicrobial activity against bacteria *E. coli*, *S. aureus,* and yeast *C. albicans*. The cytotoxicity of polypropylene composites was evaluated and proven not to be cytotoxic to human keratinocyte cells (HaCaT). Polyester composites showed a 30% higher content of Cu on their surface when compared to polyamide composites, thus showing better antibacterial and fungistatic activities. Heliopoulos et al. also reported work combining Cu and sodium alginate in viscose fabrics. Here, alginate was used as a Cu uptake booster into the fabric and as a stabilizer. Alginate-treated fabrics showed excellent antibacterial activity against Gram-negative cyanobacterium *Synechocystis* sp., increased Cu loading by 145%, prevented the loss of Cu and antibacterial protection after 50 washing cycles, and increased the UV-protective factor (UPF) by 260% when compared to viscose fabrics without alginate treatment [93].

### 2.3. Starch

Starch is a natural polymer of particular interest for numerous industrial applications due to its promising physicochemical characteristics, such as biocompatibility, biodegradability, non-toxicity, and cohesive film-forming properties. This renewable biopolymer is one of the least expensive polysaccharides. It is abundantly available and can be extracted from different parts of plants, such as stalks, roots, and seeds, with the main sources being cassava, wheat, rice, corn, and potatoes [95,96]. Chemically, starch is a semicrystalline polymer of anhydroglucose units linked by α-(1,4)-glycosidic bonds, and it is composed of two monomers: amylose and amylopectin. Amylose consists of a linear glucose chain and is responsible for the amorphous structure in starch granules, representing 15–30% of their composition. In contrast, amylopectin is a branched glucose chain with crystalline zones and represents 70–85% of the starch [97,98].

As a versatile polymer, starch is used in a variety of industrial applications, including food, paper, adhesives, paints, coatings, pharmaceuticals, and textiles [99]. Regarding the textile sector, starch is widely used as a bio-based sizing agent in textiles due to its satisfactory adhesion to fibers and good film properties, which protect yarns from mechanical abrasion [100,101].

In addition to its application in the production of traditional textiles, starch also has potential for the development of functional textiles, namely in functionalization with MNPs. In the green synthesis of MNPs, starch can act as a surface-capping agent; as a complexing agent to stabilize NPs, preventing their agglomeration; or as an assistant in heterogeneous nucleation [102]. Tahereh et al. used starch as a capping agent in the in situ synthesis of ZnONPs on cotton fabrics. The results showed that starch affects the size and shape of NPs; NPs were spherical with an average diameter of 52 nm without starch and rod-shaped with an average size of 88 nm in the presence of the capping agent. Starch also contributed to increasing the water contact angle (WCA) and the durability of the NPs to washing. Regarding antimicrobial properties, cotton fabrics functionalized with ZnONPs showed the same antibacterial activity against *E. coli* with and without starch [103]. In another work, ZnONPs were also synthesized using starch (extracted from *Nypa fruticans* (Nipa)) as stabilizing agent and then applied on the outer layer of cloth face masks. The fabric coated with starch and ZnONPs exhibited antimicrobial activity against *S. aureus* and *E. coli*, and the coating endured several washings [104].

Despite being a weak reducing agent, starch can assist in nucleation, together with cellulose. An antibacterial knitted cellulose fabric was produced by impregnating AgNPs synthesized using starch as a reducing and stabilizing agent. Synthesis under autoclaving swells the starch, making the aldehyde terminal groups more available for the reduction of Ag ions. The fabric functionalized with AgNPs presented antibacterial activity against *S. aureus* and *E. coli*, and the NPs resisted several washings [105]. Starch was also used as a reducing agent in the synthesis of CuONPs, and afterward, polysaccharide sodium alginate was used to bind the CuONPs to cotton fabrics by pad–dry–cure. The obtained CuONP-coated fabrics presented hydrophobicity and antimicrobial activity against *S. aureus*, *E. coli*, *Pseudomonas fuorescens*, *B. subtilis,* and *C. albicans*, which was maintained for up to 20 washing cycles, despite a considerable reduction [106]. Biocompatible hydrogel nanocomposite fabrics with photocatalytic and antibacterial properties were developed using potassium permanganate as a cross-linking agent and precursor of manganese dioxide NPs (MnO_2_NPs). Furthermore, starch was also used as a reduction-assistant agent. Although the hydrogel depicted high antibacterial (*S. aureus* and *E. coli*) and antifungal (*C. albicans*) activity, the hydrogel nanocomposite fabrics did not show such satisfactory results due to low absorption, low loading, and poor uniformity of hydrogel nanocomposite on the fabric [107].

Notwithstanding the special functionalities offered by NPs applied to textiles, their adhesion to the substrate may be poor, limiting their efficiency over time. Furthermore, because the effects of NPs on health and the environment are still not well known, concerns may arise due to uncontrolled leaching during use, domestic washing, or disposal. To enhance the adhesion properties of cotton for the immobilization of ZnONPs, El-Nahhal et al. functionalized cotton fibers with corn starch, and then ZnONPs were synthesized in situ using ultrasound irradiation. Besides stabilizing ZnONPs by controlling their size and shape during synthesis, starch improved the content of ZnONPs loaded onto the cotton fabric. Therefore, this diminished ZnONPs leaching from the fabric surface and enhanced antimicrobial activity (*S. aureus* and *E. coli*) [108]. In the work developed by Ahmed et al., to obtain a fabric with UV-protective, self-cleaning, and flame-retardant properties, cotton fabrics were functionalized with TiO_2_NPs emulsions and a flame-retardant agent (hexamethyl triaminophosphine) by padding, using itaconic acid as an ecofriendly binding agent. Starch was used so that the itaconic acid built up a network between the starch chains and the cellulosic substrate. The network traps both TiO_2_NPs and the flame retardant agent, promoting the finishing durability [109]. Starch also has hydrogel properties, which are advantageous for the development of flame retardants. Amani et al. designed multifunctional polyester fabrics by loading starch/corn silk as natural polymers, along with in situ synthesis of ZnONPs. The ZnONPs provided antimicrobial activity and photocatalytic (self-cleaning) properties to the fabric, whereas starch imparted flame-retardance with no dripping [110]. In another work, polyelectrolyte solutions/suspensions of cationized starch and vermiculite (VMT)/TiO_2_NPs were deposited on cotton fabric through a layer-by-layer process. A hybrid system of VMT clay and TiO_2_NPs was used as anionic species and the cationized starch as cationic species to produce flame-retardant nanocomposite thin layers on cotton fabric. The obtained functionalized fabric produced with seven bilayers presented a reduction in pyrolysis of approximately 30% [111]. Table 3 presents a summary of works carried out using MNPs and starch in the development of functional textiles.

### 2.4. Cyclodextrins

Cyclodextrins (CDs) are cyclic oligosaccharides produced by the enzymatic degradation of starch and are composed of six, seven, or eight α-(1, 4)-linked α-D-glucopyranose units (α-, β-, and γ-CDs). Their structure consists of a circular and truncated cone with a hydrophobic inner cavity and a hydrophilic outer surface. The hydrophobic inner cavity can form inclusion complexes with guest molecules, which are kept together due to Van der Waals and hydrophobic forces. The most used CDs in the textile industry are β-CDs due to their simple production, availability, cavity diameter, and price. Their applications include dyeing auxiliary to improve dye adsorption and K/S; encapsulation of active substances, such as fragrances, drugs, and antimicrobial agents; and fiber spinning [112,113].

In the work developed by Keshavarz et al., a polyester fabric was modified by in situ synthesis of polyamidoamine (PAMAM)/β-CDs/Ag nanocomposites to make a fabric with antibacterial and drug-delivery properties. PAMAM allowed for the aminolysis of polyester fabric, resulting in stable linkages with β-CDs/Ag composites. The obtained nanocomposite fabric presented a drug release of the molecules loaded into the β-CDs cavities of 45% after 150 h and a microbial reduction in *E. coli*, *S. aureus,* and *C. albicans* of 100, 100, and 99%, respectively [114]. In another work, AgNPs were synthesized on β-CD/ketoconazole (KZ) composite and then loaded onto cotton fabric to create an antimicrobial drug-delivery system. Ketoconazole is an antifungal drug, and the incorporation of AgNPs on β-CDs/KZ improved its antimicrobial properties and governed its release rate. In the sample produced with 2% of Ag, the microbial reduction was 100% in *C. albicans* and *A. niger* and about 85% in *E. coli* and *S. aureus* [115]. Antimicrobial cotton fabrics were developed using β-CDs and sulfated β-CDs (S-β-CDs) to host AgNPs, forming inclusion complexes. The treatment with the derivative β-CDs-AgNP complex and crosslinked with ethylenediaminetetraacetic acid (EDTA) was found to be the most favorable method. S-β-CDs + AgNPs + EDTA had a bacterial reduction of 95% and 79% (before and after 10 washing cycles, respectively) against *S. aureus* and 95% and 77% (before and after 10 washing cycles, respectively) against *E. coli* [116]. Cotton fabrics with self-cleaning properties were also developed by synthesizing and depositing on cotton fabric Ag/TiO_2_/β-CD nanocomposites. Three methods were tested, and exhaustion depicted the best results. Cotton treated with Ag/TiO_2_/β-CD showed excellent dye-degradation efficiency and antibacterial activity against *S. aureus* of 96.8% [117].

Using a different methodology, innovative wound dressings were developed by impregnating non-woven PET with two types of polysaccharides, chitosan and CDs, both crosslinked with citric acid by a pad–dry–cure process. Prefunctionalized anionic PET textiles were then loaded with Ag sulfate and coated using layer-by-layer deposition of a polyelectrolyte multilayer (PEM) film consisting of anionic water-soluble poly-CDs and cationic chitosan. Poly-CD-functionalized textiles adsorbed a larger amount of Ag (450 μg·cm^−2^) compared to those functionalized with chitosan, resulting in a higher bacterial reduction against *S. aureus* and *E. coli*. In the textile coated by the PEM system, the Ag release slowed down without affecting antibacterial activity, offering bacterial reduction against *S. aureus* and *E. coli* [51]. The same approach was used to develop wound dressings loaded with Ag and ibuprofen to provide a dual therapy that was both antibacterial and antalgic. Non-woven PETs were pretreated using the same PEM system and then loaded with ibuprofen lysinate (IBU-L). The obtained CDs-Ag-PEM-IBU dressing released 104 μg·cm^−2^ (56%) of IBU-L after 30 min. Regarding the antimicrobial properties, PET-CD-Ag-PEM exhibited bacterial reduction against *S. aureus* and *E. coli* [118].

Electrospinning is a versatile and effective technique to produce long and continuous fibers, offering materials with high specific surface area, high length–width ratio, and porosity. Zhang et al. developed composite microfiber matrices of polyoxymethylene (POM)/β-CDs by electrospinning with well-dispersed AgNPs. The introduction of β-CDs leads to increased average fiber diameter (from 2.1 µm to 6.4 µm) and decreased roughness and porosity of the microfiber surface. AgNPs imparted catalytic properties to the fiber mats [119]. Table 4 presents a summary of works carried out using MNPs and CDs in the development of functional textiles.

### 2.5. Cellulose

Cellulose is a natural polymer of fibrous nature with nanoscale dimensions, thus referred to as nanocellulose [120]. Structurally, nanocellulose is a polysaccharide assembled from β-(1–4) linked anhydro-D-glucose units with a degree of polymerization of up to 20,000, depending on the cellulose source [121,122]. The natural sources of nanocellulose include plants, bacteria, algae, and animals [123]. Nanocellulose can be divided into three types: cellulose nanocrystals (CNCs), cellulose nanofibrils (CNFs), and bacterial nanocellulose (BNC). CNCs are obtained through chemical hydrolysis of pure or delignified cellulose derived from plants, with a needle-like or rod-like morphology. Strong acids hydrolyze the disordered or amorphous sections of cellulose, whereas the crystalline regions, being resistant to acid digestion, remain intact [124,125]. CNFs have a flexible and elongated cross-linked structure, and for their production, the raw material undergoes initial purification processes that facilitate disintegration before mechanical delamination is employed [121,126]. Contrary to the other nanocelluloses, BNC is obtained directly from bacteria in a pure form. It neither includes byproducts, such as lignin, pectin, nor hemicelluloses, simply requiring mercerization treatment with sodium hydroxide to remove bacterial cells, medium culture components, and other debris [126]. Nanocellulose-based materials possess very interesting chemical, physical, and biological properties. Its inertness and increased surface area, due to its nanosized structure, have an impact on the availability of hydroxyl groups, thus facilitating the adsorption of different ions, atoms, and molecules. Besides being generally mechanically strong, the morphological, chemical, and optical properties of nanocellulose can be tailored. Furthermore, it is an abundant and renewable material with a relatively reduced production cost [123,126].

Cellulose, in its different forms, is often used as a reducing or stabilizing agent, binder, and stiffener in textile finishing to aid the incorporation of MNPs onto the fabric, embedding features into the final product.

CNFs play an important role as support of NPs for wound dressings. A multicomponent interpenetrating polymeric network was formed consisting of CNFs/gelatin/aminated Ag (Ag-NH_2_) NPs with improved mechanical properties, self-recovery, and antibacterial activity against *S. aureus* and *P. aeruginosa*. This non-covalently crosslinked hydrogel proved to promote wound healing through controlled evaporative water loss and hemorrhage stoppage, resulting in wound-size reduction compared to CNFs alone [127]. The applicability of CNFs in electronic textiles (e-textiles), as reported by Nechyporchuk et al., demonstrated a superior quality of AgNP ink-printed circuits on woven cotton fabrics. The CNFs were applied as a coating, forcing the pigment to concentrate and settle on the surface, enhancing the conductivity of the printed circuits [128].

Moreover, CNFs can also act as a stabilizer for ZnONPs. Due to their good shielding effect, these are often used in UV-protective clothing finishing. However, as a result of their large surface area, ZnONPs tend to agglomerate when applied on the surface of modified cotton fabrics. Thus, the negative charges of the hydroxyl and carbonyl groups present on CNFs interact with Zn through electrostatic interactions, reducing NP aggregation. Additionally, due to their structural similarity, CNFs and cotton fabric create intermolecular hydrogen interactions, strengthening the adhesion of ZnONPs to cotton fabrics, potentially avoiding NP leaching during the laundering process [129]. From the earlier superhydrophobic to the more recent photocatalytic materials, self-cleaning textiles have gained increasing interest. Photocatalytic activity is based on the breakdown of dirt via reactions of oxidation and reduction in the presence of light [130]. TiO_2_ is one of the most used NPs because it possesses excellent photocatalytic activity, able to, e.g., decompose dye pollutants. NPs’ applicability in textile finishing has nonetheless demonstrated a lack of washing fastness because of the polymer used in the coatings [131,132]. To circumvent this issue, Kale et al. showed that cellulose derived from viscose was able to coat cotton fabric with TiO_2_NPs, promoting permanent stiffness, hydrophobicity, and self-cleaning properties [133].

Yunusov and colleagues reported the stabilizing and reducing properties of a cellulose derivative, carboxymethylcellulose (CMC) [134]. Carboxymethyl groups attached to the hydroxyl groups of glucose confer hydrophilic properties to this biopolymer [135]. To promote the antimicrobial activity of cotton fabrics against *Staphylococcus epidermidis* and *C. albicans*, sodium-carboxymethylcellulose (Na-CMC) was used as a reducing agent in the synthesis of AgNPs [135]. A summary of the different types of nanocellulose and their applications, together with different MNPs, are presented in Table 5.

### 2.6. Other Polysaccharides

MNPs have also been incorporated in textiles using additional polysaccharides, such as hyaluronic acid, pectin, pullulan, κ-carrageenan, and locust bean gum, among others. Several of these biopolymers were used as reducing and stabilizing agents of MNPs—in particular, pectin, dextran, and the simultaneous use of κ-carrageenan and locust bean gum [136,137,138,139,140]. These properties reduce the use of chemicals, making the textiles more ecofriendly. Pectin is a vastly abundant polysaccharide composed mostly of α-(1,4)-linked galacturonic acid homopolymer and is one of the most complex anionic heteropolysaccharides [141,142]. As part of the plant cell-wall constitution, it is readily available. Pectin has thus become a highly promising biopolymer, particularly due to its sustainability, biocompatibility, and biodegradability [141,143]. It is possible to obtain electrospun fibers of solely pectin; however, it is a highly challenging and demanding process [144]. Thus, pectin is usually used within a composite blend of electrospun functional composite textiles [141,144]. Hyaluronic acid is composed of several repetitions of glucuronic acid and *N*-acetyl-glucosamine. It is abundant in nature and is present in every mammalian species; however, it has also been identified in *Pseudomonas* bacteria, among other organisms. Hyaluronic acid is remarkably hydrophilic and readily soluble in water. Its rheological properties are uncommon, exhibiting high water-retention capability, usually described as a lubricant [145,146]. Its ubiquitous presence in the mammalian extracellular matrix has made hyaluronic acid a good candidate for functional medical materials that promote wound healing. Electrospun non-woven medical fabrics were reported that contained hyaluronic acid and AgNPs in their formulation [136,137,138]. These textiles contained hyaluronic acid to promote tissue regeneration and prevent cell adhesion. Although hyaluronic acid is highly soluble, its release from the textile was described as slow, taking approximately 1.5 to 2.5 days to release 50% [136,138]. These textiles displayed mitigation of inflammation, promoted collagen deposition, and impeded cell adhesion in tendon and peritendinous regions [136,137,138]. Carrageenans are polysaccharides extracted from Rhodophyta sea algae and are composed of alternating 3-linked β-D-galactopyranose and 4-linked α-D-galactopyranose [147]. Interestingly, carrageenans display a mechanical synergy with locust bean gum, which is extracted from the seeds of the carob tree, which is composed of galactomannans [148]. Both carrageenans and locust bean gum possess renowned gelling properties. Furthermore, they are readily available, sustainable, and biocompatible [140,148]. These polymers mostly envisage textile medical applications, such as wound dressings and indwelling devices (Table 6) due to their biocompatibility, biodegradability, and, in some cases, immunoregulatory activity. In a vast majority of applications, AgNPs are used as an antimicrobial agent to control and prevent infections when using these polymers. Antimicrobial effectiveness was ubiquitously evaluated through a zone of inhibition, which denotes the release of these NPs from the textiles. Nevertheless, acceptable levels of acute cytotoxicity were reported [137,143,149].

## 3. Safety Issues of MNPs and the Role of Polysaccharides

MNPs encompass several safety concerns regarding living organisms and the environment, which hinder their applicability. Their properties and corresponding potential toxicity profile is complex and may undergo numerous mechanisms. The main mechanisms for MNP toxicity comprise (i) direct MNP association with an organism’s cell surface, (ii) release of toxic ions that damage enzymes and genetic material, and (iii) generation of reactive oxygen species (ROS) and subsequent oxidative stress [152,153]. For living organisms, the toxicity of MNPs is commonly correlated to the size, shape, stability, agglomeration, surface charge, and chemistry [154,155]. MNP synthesis method, stability, life cycle, and the related disposal procedure are particularly important to determine their environmental impact [156]. For these reasons, the safe-by-design concept gains exponential importance once it foresees the risk assessment in the early stages of the development of MNPs, thus prematurely preventing negative impacts on living organisms and the environment [153,157,158].

Several toxicological aspects of MNPs are defined by their physicochemical properties. MNPs smaller than 10 nm have superior multifunctional properties (e.g., antimicrobial and UV protection) and grievous cytotoxicity, owing to facile permeation of smaller NPs into cells [159]. The most common shapes of MNPs are spheres, ellipsoids, cylinders, sheets, cubes, and rods, but other shapes can be obtained depending on the synthesis method. Spherical MNPs showed a low cytotoxicity profile when compared with other MNP shapes [160]. The agglomeration state also plays a vital role in their toxicity, since it can contribute to the sedimentation process and reduce the diffusion of MNPs, resulting in higher effective doses and organ deposition. When applied in materials, agglomeration can promote the non-uniform distribution of MNPs, resulting in materials with poor reproducibility and compromised properties [161,162]. The charge of MNPs presents an essential function in regulating protein binding to MNPs, cellular uptake, oxidative stress, autophagy, inflammation, and apoptosis. Charged MNPs proved more cytotoxic than neutral forms. Moreover, positively charged MNPs display higher cytotoxicity than negative alternatives of similar size [163,164,165].

Chemical reduction is the most common approach for the synthesis of MNPs. The most commonly used chemical reducing and stabilizing agents are extremely hazardous and depict carcinogenic and teratogenic effects. Chemical synthesis can also restrict the application possibilities and biocompatibility of MNPs [166,167]. Green synthesis, through precursors of biological origin, such as polysaccharides, prevents pollution and mitigates cytotoxicity during the early stages of chemical processes [168]. Successful synthesis and stabilization of MNPs using these biological macromolecules was achieved in several works by the coordination bonds between the metal salts and functional groups of polysaccharides [46,72,86,88,105,115,134]. Poor MNP adhesion in textiles is an important challenge for the textile industry, as it results in excessive leaching during washing and disposal, which should be considered prior to their application [169]. Adhesion may be hindered by the absence of covalent bonds between MNPs and the textile substrate as a result of leaching during washing, ironing, or rubbing or due to body sweat [169]. In this context, several strategies have been reported to enhance adhesion between MNPs and textile substrates using polysaccharides, as well as the controlled release of MNPs or ions. These properties are obtained by the creation of a thin layer of cladding over the particles and/or the entrapment of drug molecules in cavities [46,65,108,114,128]. Moreover, the controlled release from polysaccharide can be designed according to the porosity or degree of swelling, pH, temperature, ionic strength, and electric fields [170].

Polysaccharides are exceptional candidates to tailor most of the described properties. They can stabilize and control the size of MNPs during the synthesis process, prevent further agglomeration, and provide additional chemical stability. They possess various binding sites, which enables attachment to the metal surface, creating a stable organic-inorganic network [171]. However, despite the outstanding biocompatibility and biodegradability properties of polysaccharides per se and the apparent lack of side effects, cytotoxicity studies must be performed when associated with MNPs. On one hand, the hydrophilic groups of polysaccharides can form non-covalent bonds with tissue cells and operate in cell–cell recognition, thus improving adhesion. This consequently promotes the affinity of MNPs with cells, which may increase their toxicological effects [36]. On the other hand, they can diminish ROS generation and decrease toxicity [172]. Thus, the development of MNPs using polysaccharides and their application in textiles must be studied simultaneously throughout all development stages, since several parameters constrain their specific toxicity.

## 4. Conclusions

The use of polysaccharides plays a crucial role in the development of functional, eclectic, and ecofriendly textiles containing MNPs. The combination of polysaccharides and MNPs has impressively widened the scope of textile functionalization, promoting the generation of novel functions, stacking properties, and their enhancement by synergy or improved efficacy. Polysaccharides’ ability to reduce and stabilize MNPs in fabrics has considerably improved their activity, concentration, and washing fastness. Reduction in MNP concentration (without compromising activity) and superior washing fastness denote an important role in preventing MNP environmental contamination that should be strongly promoted. In addition, the fully sustainable nature of polysaccharides, allied to their ability to diminish (or eliminate) hazardous reduction chemical, should be explored. In summary, the existing interaction between polysaccharides and MNPs represents key features for future advanced textiles.

## Figures and Tables

**Figure 1 nanomaterials-12-01006-f001:**
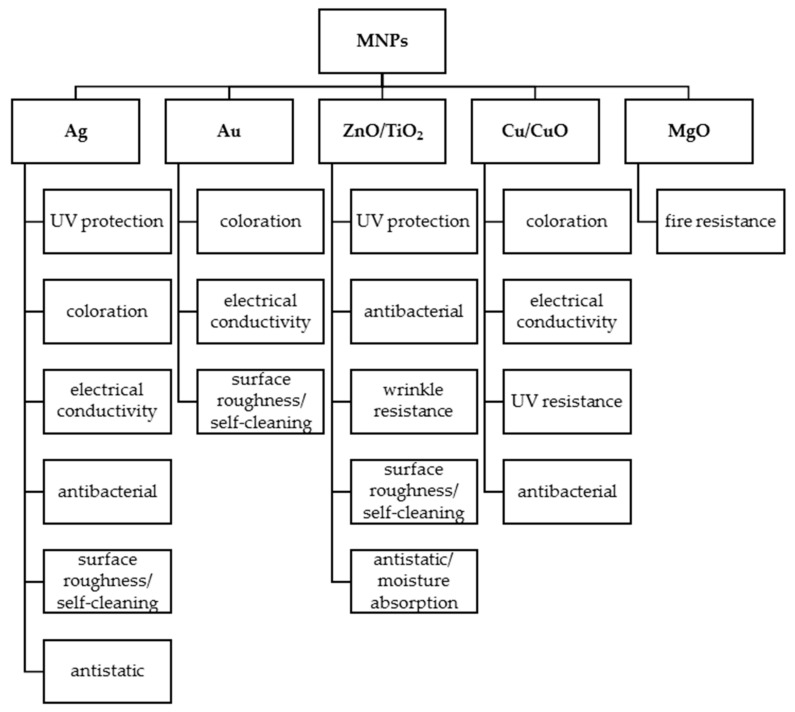
Metal and metal oxide nanoparticles used in textiles [12].

**Figure 2 nanomaterials-12-01006-f002:**
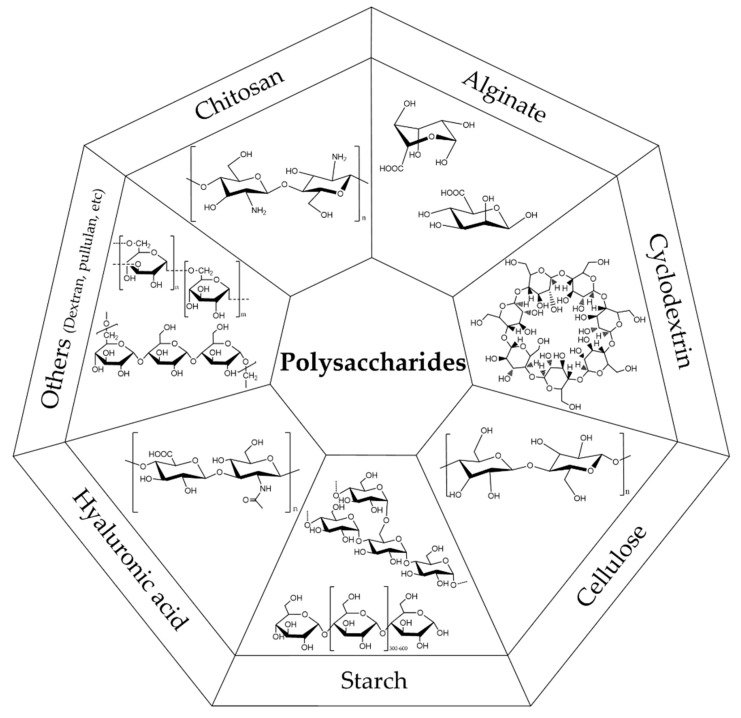
Chemical structures of polysaccharides commonly used in textile applications.

**Figure 3 nanomaterials-12-01006-f003:**
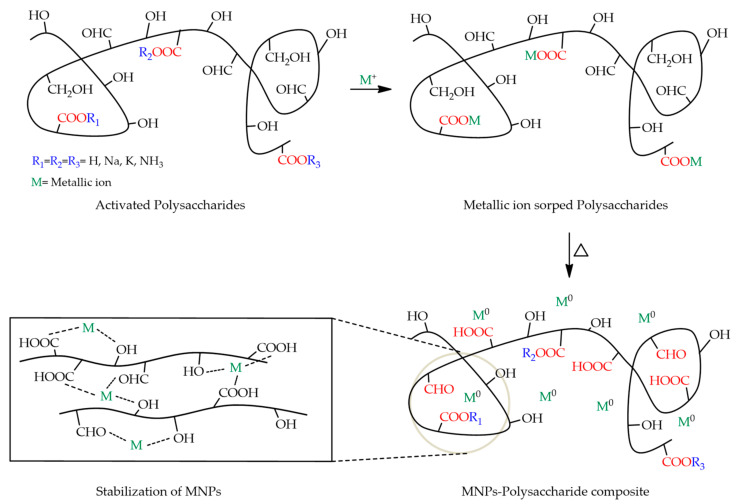
Representation of polysaccharides acting as reducers and stabilizers of metal nanoparticles.

**Table 1 nanomaterials-12-01006-t001:** Textile materials functionalized with chitosan and metal nanoparticles. Polysaccharide function towards MNPs.

Polysaccharide Function	NPs (Shape, Size)	Textile Substrate, Structure	Application	Results	Ref.
Antimicrobial activity	Ag (n.d. *)	Cotton, woven	Packaging	Antimicrobial activity against *S. aureus*, *P. aeruginosa*, *C. albicans*, and *A. niger*; chitosan increased air permeability and water absorbance	[47]
	Ag (n.d.)	Cotton, woven	Medical and UV-protective textiles	Air and water permeability decreased, whereas tensile strength and elongation increased; superior UV blocking; antimicrobial activity against *P. aeruginosa*, *S. aureus*, *A. niger,* and *C. albicans*	[48]
	Ag (spherical, <100 nm)	Cotton, woven	Hygienic products	Antibacterial activity against *S. aureus* improved with the addition of AgNPs	[49]
	Chitosan-TiO_2_ and chitosan-TiO_2_/ZnO (spherical, 11.7 nm)	Cotton, woven	Antimicrobial, self-cleaning, and UV-protective textiles	Enhanced antibacterial activity against *S. aureus*, *E. coli*, and *A. niger*; improved self-cleaning and UV-protective properties	[50]
	Ag (n.d.)	PET, nonwoven	Antimicrobial textiles (wound dressings)	Improved antibacterial activity against *E. coli* and *S. aureus*	[51]
	*Psidium guajava* extract-ZnO (spherical, 12–18 nm and 5–7 nm (water and ethanol extract)	Cotton, woven	Antimicrobial textiles	Composite with ZnONPs had better antimicrobial activity and UV protection in the presence of chitosan	[52]
	PVP-Ag (n.d., 30 nm)	Acrylic acid and chitosan-grafted polypropylene, nonwoven	Antimicrobial textiles	Antibacterial resistance increased after coating with chitosan and improved further with the addition of AgNPs (*E. coli*, *S. aureus,* and *B. subtilis*)	[53]
Antimicrobial activity; immobilization	Ag (n.d., 40–70 nm)	PET (n.d.)	Antimicrobial and flame-retardant textiles	Good antibacterial resistance against *E. coli*; flame retardance was improved with the addition of AgNPs along with chitosan	[54]
	Chitosan-Ag (spherical, 20 nm)	Cotton, woven	Antibacterial, UV-protective, and flame-retardant textiles	Antimicrobial activity against *E. coli*, *S. aureus,* and *C. albicans*; small reduction after 20 washing cycles; improvement in UV-protective and flame-retardant properties	[55]
	CMCh-Ag (spherical, 10–20 nm)	Viscose, woven	Antimicrobial textiles (wound dressings)	Superior antibacterial activity against *S. aureus* compared to that against *E. coli* with increasing concentration of AgNPs	[56]
	Ag (n.d., 34.4 nm)	Cotton, woven	Antibacterial textiles	No cytotoxic effect on human skin; excellent antibacterial durability against *E. coli* and *S. aureus* achieved by a small Ag dosage	[57]
Antimicrobial activity; immobilization; stabilizing agent	ZnO and TiO_2_ (rod-shaped, 18 nm)	Cotton, woven	Antimicrobial and UV-protective textiles	The durability of antibacterial efficiency against *K. pneumonia* and *S. aureus* increased up to 10 washing cycles the using sol–gel method	[58]
Antimicrobial activity; stabilizing agent	Chitosan-Cu (n.d., 20–30 nm)	Cotton and cotton/polyester, woven	Antimicrobial textiles	Antibacterial effect was predominantly observed against *S. aureus* in comparison with *E. coli*	[59]
Antimicrobial activity; substrate	Carboxymethyl pullulan-ZnO (spherical, 9 nm)	Cotton, woven	pH, thermo-sensitive, and antibacterial agents	Antimicrobial activity towards *S. aureus* and *E. coli;* textile sensitive to temperature between 24 and 40 °C and pH 3, 7, and 10	[14]
	Ag (n.d.)	Cotton, woven	Antimicrobial textiles	Improved antimicrobial properties against *E. coli* and *B. subtilis*	[60]
Immobilization	Ginger oil-Ag (spherical, 14 nm)	Cotton, woven	Wound patches/gauzes	Gauzes with antimicrobial activity against *C. albicans*, *E. coli*, and *S. aureus*; improved UV protection; brilliant yellow-brownish color	[17]
	Ag (n.d.)	Cotton, woven	Antimicrobial textiles, wound dressings	Good antibacterial activity against *S. aureus* and *E. coli*	[61]
	Tamarin-Ag (n.d., 20–50 nm)	Linen, woven	Antibacterial, UV-protective, and flame-retardant textiles	Antibacterial activity against *S. aureus* and *E. coli*; UV protection and improved antioxidant activity; moderate improvement of flame retardance	[62]
	Fe, Cu, Ag, Co, and Ni (n.d.)	Cotton, woven	Catalyst strips	High catalytic efficiency for the conversion of toxic substances from azo dyes and nitrophenols	[38]
	Co (n.d., 90 ± 22 nm)	Cotton, woven	Catalyst for the reduction of pollutants in water	CoNPs showed reduction of congo red dye (96% of the dye was degraded in only 21 min) and nitrophenols in aqueous solutions	[63]
	Cu (n.d., 80–90 nm)	Cotton, woven	Catalyst for dye reduction	Cu catalyst remained active even after three usages; excellent stability and recyclability during the degradation process	[64]
	ZnO and Ag (n.d., 35 and 40 nm)	Cotton, woven	Technical textiles with antimicrobial and UV protection properties	Antimicrobial action against *S. aureus* and *E. coli*; noticeable increase in UV blocking and in bending rigidity; functional properties maintained even after 15 washing cycles	[65]
	ZnO and TiO_2_ (n.d., 10–30 nm) and silicon dioxide (SiO_2_) (n.d., 10–20 nm)	Cotton/polyester, woven	Antibacterial and UV-protective textiles	Good antibacterial effect for fabrics coated with TiO_2_, followed by ZnO and SiO_2_; higher UPF for the samples with TiO_2_, followed by ZnO, SiO_2_NPs, and chitosan	[66]
	Fe (NO_3_)_3_ (n.d)	Ramie, woven	Flame-retardant textiles	Flame-retardant properties were improved; mechanical properties were reduced	[67]
Reducing and stabilizing agent	Chitosan-Ag (spherical, n.d.)	Polyamide, woven	Antimicrobial textiles	Bacterial activity with the addition of AgNPs but reduced after 20 washing cycles; consistent color, even after one year	[68]
	Chitosan-Ag (n.d.)	Sodium alginate, nanofibers	Antimicrobial textiles and filter for oil and dyes	Antibacterial effect on *E. coli* and *S. aureus*; rejection rate for oil and dye removal was significant and reduced after five filtration cycles	[69]
	Chitosan-Ag (n.d., 10–20 nm)	Polyester, woven	Coloration and antimicrobial textiles	Antibacterial activity improved but reduced after 10 washing cycles; improved color fastness	[70]
	Ag (spherical, 8.57 nm)	Viscose, woven	Antimicrobial textiles	Strong antibacterial activity against *E. coli* and *S. aureus;* tensile strength improved	[71]
Reducing and stabilizing agent; immobilization	Chitosan-Ag (spherical, n.d.)	Aramid, woven	Coloration and antimicrobial activity	Improved thermal resistivity and color properties; excellent antibacterial action against *E. coli* and *S. aureus*, even after 10 washing cycles	[46]
	Chitosan-Ag (multi-shape, 165 nm)	Cotton, woven	Antimicrobial textiles for biomedical applications	Antibacterial action against *S. aureus* and *E. coli*; coated fabric showed a higher release of Ag ions and for a longer time	[72]
Stabilizing agent	CMCh-Ag/TiO_2_ (n.d.)	Cotton, woven	Antibacterial and UV-protective textiles	Antibacterial activity against *E. coli* and *S. aureus*; UPF 50+	[73]
	Chitosan-CeO_2_ (spherical, n.d.)	Linen, woven	Antibacterial, UV protective, flame-retardant, and easy-care textiles	Antibacterial activity against *S. aureus* and *E. coli*; flame retardance was improved with the coating of chitosan and furthermore improved with the addition of CeO_2_NPs; reduced efficacy after five washes; improved wrinkle resistance and UV protection	[74]
	Ag (n.d., 63.9–68.2 nm)	Cotton, woven	Antimicrobial textiles	Antibacterial activity against *S. aureus* and *E. coli*, even after more than 50 washing cycles	[13]
	PVA-Chitosan-PEG-Ag (n.d., 96 nm)	Cotton, woven	Antibacterial nasal tampons	Reduction in blood clotting time from 180 s to 90 s and antibacterial action against *S. aureus* and *E. coli*	[44]
	Chitosan-Ag (n.d., 25 nm)	Polyamide, woven	Antimicrobial textiles (masks)	AgNPs improved antibacterial activity against *S. aureus* and *P. aeruginosa*, but it was reduced to a greater extent after washing	[75]
	CuO, ZnO, TiO_2_, and Ag (n.d., 5.8, 11.9, 15.10, and 15.93 nm)	Cotton, woven	Antimicrobial textiles	AgNPs and CuONPs showed antibacterial activity against *B. cereus* and *E. coli*, whereas ZnONPs acted against *Salmonella*, *B. cereus,* and *E. coli*	[76]
Stabilizing agent; immobilization	CMCh-Ag (spherical, 10–80 nm)	Cotton, woven	Antibacterial textiles	Improved antibacterial activity against *E. coli* and *S. aureus* before and after 50 washing cycles	[77]
Substrate	Glucose-Ag (spherical or polygon-like, n.d.)	Chitosan, non-woven	Conductive (electrocardiogram signals) and antimicrobial textiles	After eight washing cycles, the electrical resistance remained below 1 Ω·sq^−1^	[16]
	Co, Ni, Cu, and Ag (n.d., 26–33 nm)	Chitosan-TiO_2_ (<25 nm) nanofibers	Catalyst for theremoval of organic pollutants	High catalytic efficiency for the reduction of dyes and nitrophenols; good catalytic activity of Cu-composites	[78]
	CMCh-Ag/TiO_2_ (n.d., 5–15 nm)	PVA-chitosan, nanofibers	Antimicrobial textiles (wound dressings)	Antimicrobial activity against *S. aureus*, *E. coli*, *K. pneumoniae,* and *C. albicans*	[79]
Substrate; stabilizing agent	Chitosan-PVA-ZnO (n.d., 40 nm)	Chitosan-PVA-ZnO, nanofibers	Scaffolds and diabetic wound dressings	Antibacterial properties against *E. coli, P. aeruginosa, B. subtilis,* and *S. aureus*; chitosan/PVA and chitosan/PVA/ZnO nanofiber membranes with higher antioxidant properties	[45]

* n.d. corresponds to not defined.

**Table 2 nanomaterials-12-01006-t002:** Textile materials functionalized with alginate and metal nanoparticles. Polysaccharide function towards MNPs.

Polysaccharide Function	NPs (Shape, Size)	Textile Substrate, Structure	Application	Results	Ref.
Immobilization	ZnO (n.d. *)	Cotton, woven	Antimicrobial and UV-protective textiles	Enhanced UPF values and antimicrobial activity against *E. coli*, *S. aureus,* and *C. albicans*	[52]
Reducing agent; substrate	Ag (n.d.)	Alginate, electrospun fibers	Sensors	Sensitive humidity sensor for breathing monitorization (humidity range between 20% and 85%)	[15]
	Polypyrrole/Ag (n.d.)	Alginate, non-woven	Multifunctional textiles	Highly conductive, hydrophobic, and UV-resistant fabric; antistatic properties improved; thermally stable	[85]
Reducing and stabilizing agent	Ag (n.d., 6–10 nm)	Silk, woven	Multifunctional textiles	Fabric coloration; improved light and washing fastness and mechanical properties; antibacterial activity against *E. coli* and *S. aureus;* UV protection	[86]
	Ag (n.d., 8.2 nm)	Organic cotton, woven	Multifunctional textiles	Fabric coloration; washing fastness improvement; antibacterial activity against *E. coli* and *S. aureus*; UV protection	[87]
	Ag (n.d.)	Ramie, fiber	Multifunctional fibers	Fabric coloration; improved mechanical properties; antibacterial activity against *E. coli* and *S. aureus*; UV protection; reductor of 4-nitrophenol	[88]
Reducing and stabilizing agent; substrate	ZnO (rice-shaped, 100 nm)	Calcium alginate, non-woven	n.d.	Facile fabrication of ZnONPs by in situ synthesis on calcium alginate fabric	[89]
Stabilizing agent	Ag (n.d.)	Cotton gauze, non-woven	Antimicrobial textiles (wound dressing)	Excellent antibacterial efficiency against *E. coli* and *S. aureus*; improved water absorbency, water holding capacity, and vertical wicking	[61]
	SiO_2_/ZnO (spherical, 203.7 nm)	Cotton, woven	Antibacterial and UV-protective textiles	Antibacterial activity against *E. coli* and *S. aureus*; UPF 50+	[90]
	CuO and Cu_2_O (n.d., 45 and 43 nm, respectively)	Polypropylene, non-woven	Antimicrobial textiles	Excellent antimicrobial activity against *E. coli*, *S. aureus*, and *C. albicans*; non-cytotoxic to HaCaT cells	[91]
	CuO and Cu_2_O (n.d., 16–90 nm)	Polyester and polyamide, woven	Antimicrobial textiles	Excellent antimicrobial activity against *E. coli*, *S. aureus*, and *C. albicans* on polyester; good antimicrobial activity on polyamide	[92]
Stabilizing agent; immobilization	CuO (n.d.)	Viscose, woven	Antibacterial and UV-protective textiles	Excellent antibacterial activity against cyanobacterium *Synechocystis* sp.; improved UPF, washing fastness, and mechanical properties	[93]
Substrate	Ag (spherical, 10–25 nm)	Alginate, wet-spun fibers	Antibacterial textiles	Excellent antibacterial activity against *E. coli* and *S. aureus*; cytotoxic effects against cancer HeLa cells	[94]
	Ag (n.d.)	Chitosan/PET/alginate, LBL composite	Nano/ultrafiltration membranes	Antibacterial activity against *E. coli* and *S. aureus*; remotion of oils up to 93%; NP retention greater than 98%	[69]

* n.d. corresponds to not defined.

**Table 3 nanomaterials-12-01006-t003:** Textile materials functionalized with starch and metal nanoparticles. Polysaccharide function towards MNPs.

Polysaccharide Function	NPs (Shape, Size)	Textile Substrate, Structure	Application	Results	Ref.
Immobilization	ZnO (flakes and nanoflowers, 16.2 nm)	Cotton, woven	Antibacterial textile	ZnO/cotton–starch (3%) with bacterial reduction of 96% (*S. aureus*) and 76% (*E. coli*)	[108]
	ZnO (spherical, 52.42 nm); ZnO on fabric (hexagonal, 11.96 nm)	Polyester, woven	Multifunctional textiles (flame-retardant, self-cleaning, antimicrobial)	Flame-retardant with no dripping; hydrophobic with self-cleaning properties (∆RGB of 73.9); cell viability of 129%; bacteria reduction of 97%, 100%, and 94% (*E. coli*, *S. aureus*, and *C. albicans*, respectively)	[110]
	TiO_2_ (n.d. *, 200 nm)	Cotton, woven	Flame retardant	Seven bilayers: pyrolysis reduction of 30%; peak heat release rate (PHRR) of 193 W·g^−1^; Limiting oxygen index (LOI) of 22.2%	[111]
	TiO_2_ (n.d., 50–100 nm)	Cotton, linen, viscose, polyester, and their blends, woven	Multifunctional textiles (antimicrobial, self-cleaning, UV-protective)	Bacterial reduction of 85% (*S. aureus*); self-cleaning of 91%; UPF of 277 (cotton)	[109]
Reducing agent	CuO (spherical, 10–100 nm)	Cotton, woven	Antimicrobial textiles (medical, cosmetic, sports)	Hydrophobicity (WCA of 110°); antimicrobial activity of 96%, 94%, 92%, and 89% (against *S. aureus*, *E. coli*, *P. fuorescens*, *B. subtilis,* and *C. albicans*, respectively); washing durability	[106]
	MnO_2_ (n.d.)	Cotton, woven	Agriculture, medical textile, water treatment	Superabsorbent (227%); photocatalytic (∆RGB of 75); good antimicrobial properties for the hydrogel but very low for the fabric treated with the hydrogel (poor adhesion)	[107]
Reducing and stabilizing agent	Ag (n.d., 25.7 nm)	Cotton, knit	Medical textiles, water purification	Antibacterial activity against *S. aureus* and *E. coli* (halo)	[105]
Stabilizing agent	ZnO (spherical, 88 nm)	Cotton, woven	Antibacterial textiles	Hydrophobicity (WCA of 95.5°); antimicrobial activity with a zone of inhibition of 1 mm (*E. coli*); washing durability	[103]
	ZnO (n.d.)	Face masks, non-woven	Face masks	Antimicrobial activity of the ZnONPs with a zone of inhibition of 3.67 and 2.33 mm (*S. aureus* and *E. coli*, respectively)	[104]

* n.d. corresponds to not defined.

**Table 4 nanomaterials-12-01006-t004:** Textile materials functionalized with cyclodextrins and metal nanoparticles. Polysaccharide function towards MNPs.

Polysaccharide Function	NPs (Shape, Size)	Textile Substrate, Structure	Application	Results	Ref.
Reducing agent; immobilization	Ag/TiO_2_/β-CDs (semi-spherical, 48 nm)	Cotton, woven	Antibacterial textile, self-cleaning, environmental remediation	Ag/TiO_2_/β-CDs samples with excellent self-cleaning properties (methylene blue); antibacterial activity against *S. aureus* of 96.8%	[117]
	Ag (n.d. *)	PET, non-woven	Wound dressing, antibacterial, drug release	Poly-CDs: Ag adsorption of 450 μg·cm^−2^ (24 h), Ag release of 23 μg·cm^−2^ (3 days), bacterial reduction of 4 log10 (*S. aureus*) and 6 log10 (*E. coli*); PEM coating: reduced Ag diffusion (8.0 μg·cm^−2^), bacterial reduction of 3 log10 (*S. aureus*) and 5 log10 (*E. coli*)	[51]
	Ag (n.d.)	PET, non-woven	Wound dressing, antibacterial, and antalgic drug release	PEM system allowed for complete IBU-L release in 6 h; PET-CD-Ag-PEM had a bacterial reduction of 4 log10 against *S. aureus* and *E. coli*; cell viability of 0%	[118]
Reducing and stabilizing agent	β-CDs/Ag (2%) (n.d., 272.6 nm); β-CDs/KZ/Ag (2%) (n.d., 904.0 nm)	Cotton, woven	Medical applications, wound dressings, sportswear for sensitive skin	β-CD/Ag (2%): microbial reduction of 70, 42, 87, and 82% (*C. albicans*, *A. niger*, *E. coli,* and *S. aureus*, respectively); β-CD/KZ/Ag (2%): microbial reduction of 100% in *C. albicans* and *A. niger* and about 85% in *E. coli* and *S. aureus*; good washing durability (30 washing cycles)	[115]
Stabilizing agent; immobilization	Ag_2_O (n.d., 20.6 nm); Ag/β-CDs (n.d., 9.5 nm)	Polyester, woven	Drug release and antimicrobial textile	Drug release of 45% (150 h); microbial reduction in *E. coli*, *S. aureus,* and *C. albicans* of 100%, 100%, and 99%, respectively	[114]
	Ag (cubic, 31 nm)	Cotton, woven	Antibacterial textile	S-β-CDs + AgNPs + EDTA with a bacterial reduction in *S. aureus* of 95% and 79% and in *E. coli* of 95% and 77% (before and after 10 washing cycles, respectively)	[116]
	Ag (n.d.)	POM/β-CD electrospun microfiber mat	Waste treatment, molecular recognition, catalysis	Ag/POM/β-CDs mats (average fiber diameter of 6.4 μm) with excellent catalytic degradation of organic dyes in the presence of NaBH_4_	[119]

* n.d. corresponds to not defined.

**Table 5 nanomaterials-12-01006-t005:** Textile materials functionalized with cellulose and MNPs. Polysaccharide function towards MNPs.

Polysaccharide Function (Cellulose Type)	NPs (Shape, Size)	Textile Substrate, Structure	Application	Results	Ref.
Immobilization (CNFs)	Ag-NH_2_ (spherical, ~20 nm)	CNFs and gelatin, non-woven	Wound dressing	Improved mechanical, self-recovery, and hemostatic (gelation) properties; antibacterial properties against *S. aureus* and *P. aeruginosa*; fluid balance on the wound bed	[127]
	Ag (n.d. *)	Cotton, woven	Disposable e-textiles (electronic devices integrated into fabrics)	Better surface wetting and improved inkjet printing process; higher-speed inkjet printing	[128]
	ZnO (n.d., 90 ± 10 nm)	Cotton, woven	UV-protective textiles	Reduced the agglomeration of ZnO; decreased air permeability; improved mechanical properties; showed a bacteriostatic inhibition effect against *E. coli* and *S. aureus*	[129]
Immobilization (viscose)	TiO_2_ (n.d., 50 nm)	Cotton	n.d.	Photocatalytic self-cleaning and permanently stiff cotton properties; increased degradation rate of orange II dye under UV–vis light irradiation	[133]
Reducing and stabilizing agent (Na-CMC)	Ag (spherical, 2–8 nm, 5–35 nm; whiskers, L: 130–420 nm, W: 15–40 nm)	Cotton, woven	Antibacterial textiles	Bactericidal activity against bacterium *S. epidermidis* and fungus *C. albicans*	[134]

* n.d. corresponds to not defined.

**Table 6 nanomaterials-12-01006-t006:** Textile materials functionalized with other polysaccharides and metal nanoparticles. Polysaccharide function towards MNPs.

Polysaccharide Function	NPs (Shape, Size)	Textile Substrate, Structure	Application	Results	Ref.
Antimicrobial activity (Dextran)	Ag (spherical, 8–58 nm)	Cotton, n.d. *	Wound dressing	Formulations exhibited moderate antimicrobial activity against *A. niger*, *C. albicans*, *S. aureus*, and *E. coli*	[150]
Reducing and stabilizing agent (κ-carrageenan and locust bean gum)	Au (spherical, 21–45 nm)	n.d.	General use	κ-carrageenan and locust bean gum reduced and stabilized AuNPs; the formulation can be laminated on non-woven fabric at industrial large scale	[140]
Stabilizing agent (pectin)	Ag (n.d. *, 24 nm)	Pectin, PVA, PVP, and mafenide acetate, non-woven	Wound healing	Low antibacterial activity against *S. aureus*, *E. coli*, and *P. aeruginosa*; acceptable cytotoxicity, including faster in vivo wound healing	[143]
Stabilizing agent (pullulan)	Ag (spherical, 20 nm; in sodium silicate)	Cotton, n.d.	n.d.	Functionalized cotton water uptake became stimuli-responsive to pH and temperature between 24 and 30 °C (neutral and acid pH)	[151]
Substrate (pectin and hyaluronic acid)	Ag (spherical, 8.6 nm)	Pectin, hyaluronic acid, and PVA, non-woven	Wound dressing	High antimicrobial activity against *B. subtilis*, *S. aureus*, and *E. coli*; histological analysis displayed a faster healing process, attributed to the presence of hyaluronic acid	[138]
Substrate (pectin)	Ag (spherical, 3.7–8.6 nm)	Pectin, non-woven	Wound healing, catalysis, and Raman enhancement	AgNPs homogeneously distributed in the pectin nanofibers, and their size may be tailored; AgNP release took 4 weeks	[139]
Substrate (polycaprolactone and hyaluronic acid)	Ag (spherical, 4–10 nm)	Polycaprolactone and hyaluronic acid, non-woven	Prevention of post-operative tendon adhesion	Nanofiber sheath of polycaprolactone as tendon-sheet surrogate; core contains hyaluronic acid to prevent cell adhesion and AgNPs as antimicrobial agent; suitable cytotoxicity; low antimicrobial activity against *S. aureus* and *E. coli*; histological observations revealed promising antiadhesive properties	[136]
Substrate (polylactic acid and hyaluronic acid)	Ag (spherical and rods, 10–40 nm)	Polylactic acid, hyaluronic acid, non-woven	Prevention of post-operative tendon adhesion	Polylactic acid worked as a tendon-sheet surrogate, hyaluronic acid prevented cell adhesion, and AgNPs were responsible for the antimicrobial effect; most tested formulations exhibited acceptable cytotoxicity (>70%); weak antimicrobial activity against *S. aureus* and *E. coli*; in vivo tests with rats showed no blood, renal, or liver problems; histological observation denoted low adhesion in some formulations	[137]
Substrate (PVA, gum arabic, and polycaprolactone)	Ag (spherical, 10–100 nm)	PVA, gum arabic, and polycaprolactone, non-woven	Wound dressing	Low antimicrobial activity against *S. aureus*, *E. coli*, *P. aeruginosa*, and *C. albicans*. Improved adequacy of water-vapor permeability and porosity for wound-dressing use; suitable cytotoxicity	[149]

* n.d. corresponds to not defined.

## Data Availability

Not applicable.

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
