# Peer review of "Polysaccharides and Metal Nanoparticles for Functional Textiles: A Review"

_nanomaterials, 2022, doi:10.3390/nano12061006_

Round 1

Reviewer 1 Report

This review covers the use of metal based nanomaterials for the functionalization of textiles. The use of polysaccharides to facilitate the functionalization process was also reviewed. Overall, it is an interesting topic to cover that has the potential to advocate the integration of nanotechnology in the textile industry for improved applications. The authors are recommended to consider the following aspects to further improve this review. 1. Environmental sustainability and eco-friendly are topics of greater breadth that the authors failed to deliver in this review. In my opinion, it is a related topic that could be covered elsewhere. Instead, the authors should focus on the use of polysaccharides to lessen the environmental toxicity effects of metal based nanomaterials. This is highly relevant to this review and should be discussed in details. The authors are recommended to check the concept of Safer-by-design or safe-by-design, in which reducing or eliminating the metal shedding is highly important and also relevant to this review. 2. Fig 1 listed the use of different metal based nanomaterials in textile. It would be much more informative if the authors could summarize and group the materials that are applied in the same way. That would also reduce the unnecessary repetition of the same applications over and over again. 3. Fig 3 is not that useful. It's too generic and has to much to show. 4. Similarly, Table 1 is a mess. The authors could have group the polysaccharides by types, or group the namomaterials, or group the applications. The rest of tables has the same problem.

Author Response

Dear Dr. Katarina Nesovic, Assistant Editor of Nanomaterials,

We would like to acknowledge the Editor and all the Reviewers for such a careful and insightful review. It is plainly clear that the Reviewers performed a thorough and interested reading of our work, for which we are extremely grateful. We carefully analysed all comments and did our best to meet the Reviewers expectations. We are well aware that if we did so we would considerably improve the clarity, quality and impact of the manuscript.

Therefore, all Reviewers comments were carefully analysed and the responses are provided after each comment. The revised manuscript contains all performed track changes.

Reviewer #1

This review covers the use of metal based nanomaterials for the functionalization of textiles. The use of polysaccharides to facilitate the functionalization process was also reviewed. Overall, it is an interesting topic to cover that has the potential to advocate the integration of nanotechnology in the textile industry for improved applications.

The authors would like to thank the Reviewer for such a positive comment, and for finding our work relevant and impactful.

Comment #1: Environmental sustainability and eco-friendly are topics of greater breadth that the authors failed to deliver in this review. In my opinion, it is a related topic that could be covered elsewhere. Instead, the authors should focus on the use of polysaccharides to lessen the environmental toxicity effects of metal based nanomaterials. This is highly relevant to this review and should be discussed in details. The authors are recommended to check the concept of Safer-by-design or safe-by-design, in which reducing or eliminating the metal shedding is highly important and also relevant to this review.

Response #1: The authors acknowledged the highly interesting comment, and we fully agree with the Reviewer. We added information concerning the impacts of the environmental toxicity of metal based nanomaterials, as suggested by the Reviewer. Therefore, a novel subsection was added to the text (from line 641 to 703):

“3. Safety issues of MNPs and the role of polysaccharides

MNPs encompass several safety concerns regarding living organisms and the envi-ronment, which hinder their applicability. Their properties and corresponding potential toxicity profile is complex and may undergo numerous mechanisms. The main mecha-nisms for MNPs toxicity comprise i) direct MNPs association with an organism’s cell surface, ii) release of toxic ions that damage enzymes and genetic material, and iii) gener-ation of reactive oxygen species (ROS) and subsequent oxidative stress [153]. For living organisms, the MNPs’ toxicity is commonly correlated to the size, shape, agglomeration state, surface charge, stability, and surface chemistry [154,155]. MNPs synthesis method, stability, life cycle, and the related disposal procedure are particularly important to deter-mine their environmental impact [156]. For these reasons, the safe-by-design concept is gaining exponential importance once it foresees the risk assessment in the early stages of the development of MNPs. Thus, prematurely preventing negative impacts on   living organisms and the environment [157].

Several toxicological aspects of the MNPs are defined by their physicochemical prop-erties. The size of MNPs below 10 nm is related to superior multifunctional properties (e.g., antimicrobial and UV protection) and grievous cytotoxicity owed to facile permeation of smaller NPs into cells [158]. The most common shapes of MNPs are spheres, ellipsoids, cylinders, sheets, cubes, and rods, but other shapes can be obtained depending on the synthesis method. Spherical MNPs showed a low cytotoxicity profile when compared with other MNPs shapes [159]. The agglomeration state also plays a vital role in their tox-icity, since it can contribute to the sedimentation process and reduce the diffusion of MNPs, resulting in higher effective doses and organ deposition. When applied in materi-als, the agglomeration can promote the non-uniform distribution of MNPs, resulting in materials with poor reproducibility and compromised properties [160,161]. The charge of MNPs presents an essential function in regulating the protein bindings to MNPs, cellular uptake, oxidative stress, autophagy, inflammation, and apoptosis. Charged MNPs showed to be more cytotoxic than neutral forms. Moreover, positively charged MNPs de-pict higher cytotoxic than negative alternatives of similar size [162-164].

Chemical reduction is the most common approach for the synthesis of MNPs. The most commonly used chemical reducing and stabilizing agents are extremely hazardous and depict carcinogenic and teratogenic effects. The chemical synthesis can also restrict the application possibilities and biocompatibility of the MNPs [165,166]. Green synthesis, through precursors of biological origin such as polysaccharides, prevents pollution and mitigates cytotoxicity during the early stages of chemical processes [167]. The successful synthesis and the stabilization of MNPs using these biological macromolecules was achieved in several works by the coordination bonds between the metal salts and func-tional groups of polysaccharides [46,72,87,89,106,116,135]. Regarding the application of MNPs in textiles the main challenges and disadvantages are related to poor adhesion to textile substrates, leaching problems during washing and disposal, which should be con-sidered before application [168]. Adhesion may be hindered by the absence of covalent bonds between the MNPs and the textile substrate, being leached during washing, ironing, rubbing, or due to body sweat [168]. In this context, several strategies have been reported to enhance the adhesion between MNPs and textile substrates using polysaccharides, as well as the controlled release of MNPs or ions. These properties are obtained by the crea-tion of a thin layer of cladding over the particles and/or the entrapment of the drug mole-cules into cavities [46,65,109,115,129]. Moreover, the control release from polysaccharide can be designed according to the porosity or degree of swelling, pH, temperature, ionic strength and electric fields [169].

Polysaccharides are exceptional candidates to tailor most of the described properties. They can stabilize and control the size of MNPs during the synthesis process, prevent further agglomeration and provide additional chemical stability. They possess various binding sites, which enable the attachment to the metal surface, creating a stable organ-ic-inorganic network [170]. However, despite the outstanding biocompatibility and bio-degradability properties of polysaccharides per se and the apparent lack of side effects, cy-totoxicity studies must be performed when associated with MNPs. On one hand, the hy-drophilic groups of polysaccharides can form non-covalent bonds with tissue cells and operate in cell-cell recognition, thus improving adhesion. This consequently promotes the affinity of MNPs with cells, which may increase their toxicological effects [36]. On the oth-er hand, they can diminish ROS generation and decrease the toxicity [171]. Thus, the de-velopment of MNPs using polysaccharides and their application in textiles must be stud-ied simultaneously throughout all development stages since several parameters constrain their specific toxicity.”

Comment #2: Fig 1 listed the use of different metal based nanomaterials in textile. It would be much more informative if the authors could summarize and group the materials that are applied in the same way. That would also reduce the unnecessary repetition of the same applications over and over again.

Response #2: Thank you for your observation. We altered Figure 1 to make it more reader friendly, by eliminating unnecessary repetitions.

Comment #3: Fig 3 is not that useful. It's too generic and has too much to show.

Response #3: We understand the Reviewers point of view, however, allow us to disagree. In our opinion, Figure 3 is meant to be generic to encompass all mentioned polysaccharides. Therefore, in our understanding, Figure 3 depicts in a reader friendly way the chemistry behind the reduction process, without specifying any of the polysaccharides. Therefore, an additional explanation was inserted in the text addressing Figure 3 (from line 95 to 105): “As depicted in Figure 3, polysaccharides host the metal ions through noncovalent bonding (sorption). The as formed metallic precursor is then reduced to a zero-valent state, starting nucleation and nanocrystal growth, just by altering the order of free energy (heating). The increase in temperature will stabilize the MNPs and allow the control of their morphology and growth kinetics. This type of self-assembly (bottom-up) synthesis is preferred over top-down synthesis, where the starting materials are reduced in size via mechanical, thermal, or chemical treatments. These treatments may induce an unwanted oxidation of the NPs, and consequently alter their physical properties and/or surface chemistry. Furthermore, the stabilized MNPs will not easily leach the coordinated metal ion unless there is an external stimulus, such as a pH change. Most polysaccharides are susceptible to pH alteration and thus often used for controlled release and drug delivery in polysaccharide-based systems”. The following reference was added to support these statements:

Wang, C.; Gao, X.; Chen, Z.; Chen, Y.; Chen, H. Preparation, Characterization and Application of Polysaccharide-Based Metallic Nanoparticles: A Review. Polymers 2017, 9, 689. https://doi.org/10.3390/polym9120689

Comment #4: Similarly, Table 1 is a mess. The authors could have group the polysaccharides by types, or group the namomaterials, or group the applications. The rest of tables has the same problem.

Response #4: We agree with the Reviewer comment; thus we have revised the tables accordingly. Now, the first column of every table displays the polysaccharide function towards MNPs. The tables captions were also revised. (Please see revised manuscript).

Reviewer 2 Report

The review entitled “Polysaccharides and Metal Nanoparticles for Functional Textiles: A Review” by  Jorge Padrão et al (Manuscript ID: nanomaterials-1616058) is well structured and easy to follow. It is publishable after minor revisions. Further review is not needed and I commend the authors for this work certainly very helpful and useful for researchers that want to enter the field of functional textiles.

My requests are listed below.

1) Introduction, page 2, line 50: among the metal mentioned there is also gold (Au) however this metal is never mentioned in the rest of the review. Can the authors explain the reason?

2) The reduction mechanism depicted in Figure 3 should be explained better within the text.

3) Some abbreviations and acronyms are not explained (i.e.: UPF, K/S ecc.).

4) Page 8, line 135: replace Ni+ with Ni2+ (the same mistake is also present in the original paper).

Author Response

Dear Dr. Katarina Nesovic, Assistant Editor of Nanomaterials,

We would like to acknowledge the Editor and all the Reviewers for such a careful and insightful review. It is plainly clear that the Reviewers performed a thorough and interested reading of our work, for which we are extremely grateful. We carefully analysed all comments and did our best to meet the Reviewers expectations. We are well aware that if we did so we would considerably improve the clarity, quality and impact of the manuscript.

Therefore, all Reviewers comments were carefully analysed and the responses are provided after each comment. The revised manuscript contains all performed track changes.

Reviewer #2

The review entitled “Polysaccharides and Metal Nanoparticles for Functional Textiles: A Review” by Jorge Padrão et al (Manuscript ID: nanomaterials-1616058) is well structured and easy to follow. It is publishable after minor revisions. Further review is not needed and I commend the authors for this work certainly very helpful and useful for researchers that want to enter the field of functional textiles.

The authors are grateful for so many positive comments to our work by the Reviewer.  

Comment #1: Introduction, page 2, line 50: among the metal mentioned there is also gold (Au) however this metal is never mentioned in the rest of the review. Can the authors explain the reason?

Response #1: Thank you for your comment, but please allow us to disagree. There are only one direct mention to gold (Au), thus it is easy to miss. Gold (Au) nanomaterials are depicted in Table 6 (Reference 141), which refer the use of k-carrageenan and locust bean gum for the reduction and stabilization of AuNPs, respectively. However, despite our thorough review, this was the only reference found mentioning the use of gold nanomaterials in combination with polysaccharides for textile applications in the last 5 years. The authors are well aware of the common use of gold nanoparticles in textiles applications (please see reference [7]), in particular for drug delivery. However, they are not combined with polysaccharides. Therefore, we could not consider them in our work.

Comment #2: The reduction mechanism depicted in Figure 3 should be explained better within the text.

Response #2: We acknowledge and fully agree with the Reviewer comment, therefore we added information in the text to further explain Figure 3 (from line 94 to 105): “As depicted in Figure 3, polysaccharides host the metal ions through noncovalent bonding (sorption). The as formed metallic precursor is then reduced to a zero-valent state, starting nucleation and nanocrystal growth, just by altering the order of free energy (heating). The increase in temperature will stabilize the MNPs and allow the control of their morphology and growth kinetics. This type of self-assembly (bottom-up) synthesis is preferred over top-down synthesis, where the starting materials are reduced in size via mechanical, thermal, or chemical treatments. These treatments may induce an unwanted oxidation of the NPs, and consequently alter their physical properties and/or surface chemistry. Furthermore, the stabilized MNPs will not easily leach the coordinated metal ion unless there is an external stimulus, such as a pH change. Most polysaccharides are susceptible to pH alteration and thus often used for controlled release and drug delivery in polysaccharide-based systems”. The following reference was added to support these statements:

Wang, C.; Gao, X.; Chen, Z.; Chen, Y.; Chen, H. Preparation, Characterization and Application of Polysaccharide-Based Metallic Nanoparticles: A Review. Polymers 2017, 9, 689. https://doi.org/10.3390/polym9120689

Comment #3: Some abbreviations and acronyms are not explained (i.e.: UPF, K/S ecc.).

Response #3: The authors would like to thank the Reviewer for such a careful reading of our work. The K/S abbreviation definition is present in line 306, however we missed the definition of UPF acronym. This was revised in line 393 and 394: “ultraviolet protective factor (UPF)”.

Comment #4: Page 8, line 135: replace Ni+ with Ni2+ (the same mistake is also present in the original paper).

Response #4: This typo was revised. Again, the authors would like to acknowledge the thorough review.

Round 2

Reviewer 1 Report

The authors have improved the quality of the manuscript in this revision. Minor suggestions are as follows,

  1. The "safe-by-design" strategy was brought up by others previously. Although it did not specifically talk about the role of polysaccharides, the authors should consider citing a few of the most important literatures in the field, such as Adv Mater 2019, 31, 1805391; Adv Mater 2018, 30, 1705691.
  2. The writing could be improved by a thorough read-through to remove typos and inconsistencies.

Author Response

Reviewer #1

  1. The "safe-by-design" strategy was brought up by others previously. Although it did not specifically talk about the role of polysaccharides, the authors should consider citing a few of the most important literatures in the field, such as Adv Mater 2019, 31, 1805391; Adv Mater 2018, 30, 1705691.

R.1 The authors would like to thank the Reviewer for such interesting references. In our opinion, these works fall within the scope of the manuscript and were added:

Reference: Adv Mater 2019, 31, 1805391 – line 652 and line 659 – Reference [154], highlighted in yellow.

Reference: Adv Mater 2018, 30, 1705691 – line 659 – Reference [159], highlighted in yellow.

  1. The writing could be improved by a thorough read-through to remove typos and inconsistencies.

R2. The Reviewer was correct. We preformed several minor corrections as clearly observable in the manuscript. Thank you.
